# IL-13Rα2 Is a Biomarker of Diagnosis and Therapeutic Response in Human Pancreatic Cancer

**DOI:** 10.3390/diagnostics11071140

**Published:** 2021-06-23

**Authors:** Toshio Fujisawa, Bharat H. Joshi, Sho Takahashi, Yusuke Takasaki, Akinori Suzuki, Koichi Ito, Kazushige Ochiai, Ko Tomishima, Shigeto Ishii, Raj K. Puri, Hiroyuki Isayama

**Affiliations:** 1Department of Gastroenterology, Graduate School of Medicine, Juntendo University, Tokyo 113-8421, Japan; t-fujisawa@juntendo.ac.jp (T.F.); sho-takahashi@juntendo.ac.jp (S.T.); ytakasa@juntendo.ac.jp (Y.T.); suzukia@juntendo.ac.jp (A.S.); kitoh@juntendo.ac.jp (K.I.); kaz.ochiai.0728@gmail.com (K.O.); tomishim@juntendo.ac.jp (K.T.); sishii@juntendo.ac.jp (S.I.); 2Tumor Vaccines and Biotechnology Branch, Division of Cellular and Gene Therapies, Center for Biologics Evaluation and Research, Food and Drug Administration, Silver Spring, MD 20993, USA; Bharat.Joshi@fda.hhs.gov

**Keywords:** IL-13R**α**2, cancer invasion, metastasis, pancreatic cancer, AP-1 pathway

## Abstract

IL-13Rα2 is a high-affinity binding protein for its ligand IL-13 and a cancer-testis antigen as it is expressed in the testis. IL-13Rα2 is highly expressed in various cancers, including pancreatic cancer, and consists of three domains: extracellular, transmembrane, and cytoplasmic. The extracellular domain binds to the ligand to form a biologically active complex, which initiates signaling through AP-1 and other pathways. IL-13Rα2 is also expressed in diseased cells such as fibroblasts that are involved in various inflammatory diseases, including cancer. We have reported that IL-13Rα2 is a prognostic biomarker for malignant glioma, adrenocortical cancer, and pancreatic cancer. In pancreatic cancer, a small sample of tissue could be examined for the expression of IL-13Rα2 by using the endoscopic ultrasound-fine needle aspiration technique (EUS-FNA). In addition, a peptide-based targeted approach using Pep-1L peptide could be used to study the biodistribution and whole-body cancer imaging for the screening of pancreatic cancer in suspected subjects.

## 1. Structure and Signal Transduction through IL-13R in Pancreatic Cancer

IL-13 binds to two primary chains, IL-13Ra1 and IL-13Ra2, and together with IL-4Rα form three different receptor configurations [1,2]. Type I IL-13R consists of the IL-13Rα1, IL-13Rα2, and IL-4Rα chains. This type of configuration is expressed in a variety of solid human tumor cells such as glioma, pancreatic cancer, head and neck cancer, and ovarian cancer [3,4]. IL-13 binds to IL-13Rα1 and recruits IL-4Rα to form a functional receptor complex. IL-13 can also bind to IL-13Rα2 with high affinity and does not seem to require other chains for signaling [5]. Type II IL-13R consists of the IL-13Rα1 and IL-4Rα chains. Endothelial cells, fibroblast, and some tumor cells express this receptor type [6]. Type III IL-13R includes not only IL-13Rα1 and IL-4Rα, but also the IL-2RγC chain. Although IL-13 does not bind IL-2RγC, IL-2RγC can interfere with IL-13 binding and signaling when it is introduced into cells that naturally do not express this chain.

After forming a complex with IL-13R**α**2, IL-13 participates in inflammation, allergy, and immune regulation [7,8]. Moreover, IL-13 plays a key role in immune-related diseases such as asthma, pulmonary fibrosis, and ulcerative colitis; therapies for these diseases targeting IL-13 are being explored [9]. The classical signaling pathway of IL-13 activation is JAK/STAT6 via type I and type II IL-13 receptor complexes [10,11]. IL-13Rα2 was once believed to be a decoy receptor because its intracellular domain was too short to transmit signal to downstream molecules. However, IL-13Rα2 can signal through the AP-1 pathway. This contradictory phenomenon originates from the existence of two isoforms of IL-13Rα2. The first isoform is an extracellular domain that saturates the IL-13 ligand, thus behaving like a decoy receptor and inhibiting the effects of IL-13 [12]. The second isoform participates in signal transduction when it binds to IL-13 and cooperates with other molecules such as Chitinase 3-like 1 (CHI3L1) [13] and the epidermal growth factor receptor variant III (EGFRvIII) [14]. These two isoforms compete with each other for binding to the same ligand.

Although IL-13Rα1 is expressed at basal levels and uniformly in a variety of cancer and non-cancer cells and tissues, IL-13Rα2 is predominantly overexpressed in a variety of solid cancers and inflammatory pathologies. The overexpression of IL-13Rα2, particularly in certain types of cancers, has led to the hypothesis that it may serve as a signature gene and a target for cancer detection and therapy. 

## 2. Usefulness of IL-13Rα2 for the Diagnosis of Pancreatic Cancer

IL-13Rα2 is highly expressed in various solid cancers, including pancreatic cancer [15], glioblastoma [16], ovarian cancer [4,17], breast cancer [18], colon cancer [19], prostate cancer [20], melanoma [21], and so on [22,23]. In pancreatic cancer, we have demonstrated that approximately 50% of primary and established cell lines express high levels of IL-13Rα2 [24], and more than 70% of clinical specimens express high levels as well [3,25]. IL-13 Rα2 is expressed specifically in tumor cells within the tumor and is absent or weakly expressed in the surrounding stroma. Similarly, it has been reported that the high expression of IL-13Rα2 is observed in 100% of pheochromocytoma [22], 66% of colorectal cancer [26], 62% of glioma [14], 50% of gastric cancer [27], and 7.5% of melanoma [21]. As IL-13 Rα2 is heterogeneously expressed in different types of tumors, clinical stage, and pathological grades, its expression levels vary within a cancer entity or across different cancer.

Recently, we have demonstrated that IL-13Ra2 is expressed in invading tumor cells in the perineural invasion of PDA and its expression is associated with the histologic type and the clinical stage of the disease [25]. The expression of IL-13Rα2 in cancer tissues is commonly evaluated by quantitative or semi-quantitative immunohistochemistry (IHC), and less commonly by in situ hybridization (ISH) [25]. Rarely, methods such as electrochemical immunosensor [19] and quantitative proteomics [28] are used for evaluation. IHC is currently a gold standard for the evaluation of the expression of antigens and receptors in clinical samples including IL-13Rα2 expression in PDAC. Although IHC results were confirmed by in-situ hybridization (ISH) technique, it remained quite challenging to determine sensitivity and specificity of IHC as this technique is subjective. We used isotype control antibodies for specificity and counted % positive cells in each sample for sensitivity in a blinded manner. Our future aim is to include additional methods/technologies such as real time-PCR and ELISA to corroborate IHC/ISH results for additional sensitivity. We evaluated the expression of IL-13Rα2 in surgically resected pancreatic cancer and determined any association with post-operative patient prognosis. There were more post-operative tumor recurrences, and survival of subjects was significantly shortened in the group with high IL-13Rα2 expression. Although these differences in expression levels associated with histologic types have not yet been analyzed on a large set of clinical specimens, IL-13Rα2 expression may serve as a biomarker of disease activity and as a predictor of disease relapse.

Because the histologic diagnosis of pancreatic cancer is not directly accessible to the endoscope, the endoscopic ultrasound-guided fine-needle aspiration (EUS-FNA) is important for histo-pathological examination. Tissue sampling with EUS-FNA can sometimes yield a small sample, in which cell-surface antigen expression is useful for confirming the diagnosis [29]. However, IL-13Ra2 expression is heterogenous in most cancers and because of the heterogeneity in expression, it is possible one will miss samples by EUS-FNA. Therefore, it is recommended to sample at various locations of tumors. We are currently investigating whether the expression of IL-13Rα2 can be confirmed preoperatively using EUS-FNA samples. 

IL-13Rα2 may also contribute to imaging for cancer detection. Sai et al. radiolabeled Pep-1L, a peptide isolated from a hepta-peptide library that specifically binds to IL-13Rα2, with ^64^Copper designed for tumor imaging and subsequently examined the effectiveness of positron emission tomography/computed tomography (PET/CT). In orthotopic mouse models of glioblastoma and malignant melanoma, they showed that [^64^Cu]Pep-1L selectively binds to IL-13Rα2-expressing G48 tumors revealing intra-cranial IL-13Rα2-expressing glioblastoma [30]. A similar technique could be utilized for the diagnosis of pancreatic cancer expressing IL-13Rα2.

## 3. Biological Functions of IL-13Rα2 in Pancreatic Cancer

IL-13Rα2 is localized mainly on the tumor cell surface and its expression is positively corelated with the severity of the disease, including an advanced tumor stage and the pathological grade, suggesting that IL-13Rα2 may have a critical role in tumor progression. We evaluated the relationship between IL-13Rα2 expression and tumor progression in animal models. In an orthotopic pancreatic cancer mouse model, the IL-13Rα2-positive tumor was highly invasive to surrounding organs and cause metastases. In contrast, the gene silencing of IL-13Rα2 by siRNA technology ameliorated tumor invasion and metastasis [15]. Thus, in vivo mouse models of human PDA indicate that IL-13Ra2 is one of the key-signature genes participating in tumor invasion and metastasis. In another way, the upregulation of IL-13Rα2 by plasmid vector in the IL-13Rα2-negative tumor resulted in enhanced tumor progression. Survival time in the mice with IL-13Rα2-positive tumor was significantly shorter than the mice with IL-13Rα2-negative tumors [15].

## 4. Mechanisms by Which IL-13Rα2 Promotes Cancer Invasion and Metastasis

We have investigated the mechanisms of cancer invasion and metastasis as well as the involvement of the IL-13/IL-13Rα2 axis in pancreatic cancer. We used two human pancreatic cancer cell lines: HPAF-II, which is IL-13R**α**2-negative, and HS766T, which is IL-13Rα2-positve. We demonstrated that IL-13Rα2 promoted pancreatic cancer metastasis by activating matrix metalloproteinases (MMPs) transcription, and its effect was reversed by the knockdown of IL-13Rα2 expression [15]. Conversely, the knock-in of IL-13Rα2 to IL-13Rα2-negative pancreatic cancer cells increased tumor invasion and metastasis. Examining the pathways that promote the transcription of MMPs revealed that IL-13Rα2 increased the transcription of MMPs by activating extracellular signal-regulated kinase (ERK) 1/2 and stimulating the action of the transcription factor, activator protein-1 (AP-1) including c-jun and c-fos (Figure 1). 

In other cancer types, the mechanisms and signal transduction of IL-13Rα2 have also been reported. The signaling pathway through IL-13Rα2 in glioblastoma multiforme was also examined in our laboratory [31]. Similar to pancreatic cancer, two AP-1 transcription factors, c-jun and Fra-1, played an important role in the IL-13Rα2 signaling in human glioma samples. Moreover, IL-13Rα2 activates src, phosphatidylinositol 3 kinase (PI3K), Akt, and mTOR in glioma cells [32]. Src suppression attenuates the activation of the src/PI3K/Akt/mTOR pathway induced by IL-13Rα2. Papageorgis et al. reported that IL-13Rα2 silencing enhanced the phosphorylation of STAT6 and impaired the migratory ability of metastatic breast cancer cells [33]. Moreover, genome-wide transcriptional analysis revealed that IL-13Rα2 knockdown upregulated the metastasis of tumor-suppressor protein 63 (TP63) in a STAT6-dependent manner. The roles of IL-13R**α**2 in pancreatic cancer are summarized in Table 1.

## 5. Mechanisms of the Increased Expression of IL-13Rα2 in Pancreatic Cancers

We have investigated the biological implications of IL-13Rα2 overexpression in pancreatic cancer. Epigenetic regulation of IL-13Rα2 was examined and the copy number variation and mutation of cDNA in the IL-13Rα2 gene were examined. There is only one CpG site in the IL-13Rα2 promoter region, but that CpG site was not methylated in two pancreatic cancer cell lines and in the normal epithelial cells of the pancreatic duct [24]. However, histones in IL-13Rα2 promoter regions were highly acetylated. Interestingly, when cells were treated with histone deacetylase (HDAC) inhibitors, not only histone acetylation but also IL-13Rα2 expression was dramatically enhanced in IL-13-receptor negative pancreatic cancer cells. Moreover, c-jun in IL-13Rα2-positive cells was expressed at higher levels compared to IL-13Ra2-negative cells, and two kinds of c-jun inhibitors prevented the increase of IL-13Rα2 by HDAC inhibition. These results suggest that histone acetylation and c-jun expression are important factors in the increased expression of IL-13Rα2 in pancreatic cancer (Figure 2). The mechanism for the control of IL-13Rα2 expression is shown in Table 1.

## 6. Future Diagnostic and Prognostic Trends for Pancreatic Cancer in the Context of IL-13Rα2 Expression

When the usefulness of PET/CT targeting IL-13Rα2 is established, it may be more useful for specific diagnoses of cancer compared to FDG-PET, as the latter reflects the cancer metabolism attribute of the tumor. PET/CT targeting IL-13Rα2 may be useful not only for the diagnosis of pancreatic cancer but also evaluating the expression level of IL-13Rα2 and predicting the efficacy of therapy targeting IL-13Rα2. In addition, extracellular domain of IL-13Rα2 may be cleaved and shed in the systemic circulations. Therefore, serum marker by liquid biopsies could be useful as potential biomarker of diagnosis and prognosis and possibly predicting the efficacy of therapy targeting IL-13Rα2. Animal studies and clinical studies are needed to determine whether IL-13Ra2 is secreted in the serum and does it correlate with the clinical outcome. Somatostatin receptor scintigraphy is used not only for the qualitative diagnosis of pancreatic neuroendocrine tumors but also for the pre-treatment evaluation of peptide receptor radionuclide therapy by evaluating somatostatin receptor expression.

The results of our study using resected pancreatic cancer specimens suggest a relationship between IL-13Rα2 expression and perineural invasion [25]. In IL-13Rα2-positive pancreatic cancer, the possibility of celiac plexus invasion can be predicted in advance of surgery, and it may be possible to manage the pain in the patients receiving chemotherapy. Raza et al. developed an anti-IL-13Rα2-based novel therapy for assessing its antitumor effect in a mouse model of lung cancer [34]. The anti-IL-13Rα2 antibody reduced tumor metastasis and prolonged the prognosis in this model. In addition, the combinatorial approach of using an IL-13 neutralizing monoclonal antibody [35] and a blocking peptide to IL-13Rα2 [36] were investigated for antitumor effects in mouse models. Okamoto et al. found that the expression of amphiregulin, which is a member of the epidermal growth factor (EGF) family, positively correlated with IL-13Rα2 expression and promoted tumor growth via the activation of amphiregulin in melanoma [21].

In conclusion, IL-13Rα2 can be a potential diagnosis and prognostic biomarker and a marker for therapeutic response to conventional as well as cutting-edge cellular and gene therapy approaches for pancreatic cancer.

## Figures and Tables

**Figure 1 diagnostics-11-01140-f001:**
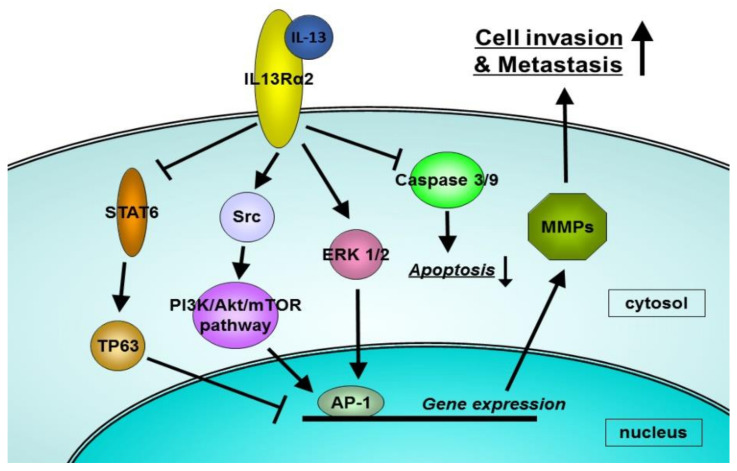
The schema of mechanisms by which IL-13Rα2 promotes cancer invasion of metastasis.

**Figure 2 diagnostics-11-01140-f002:**
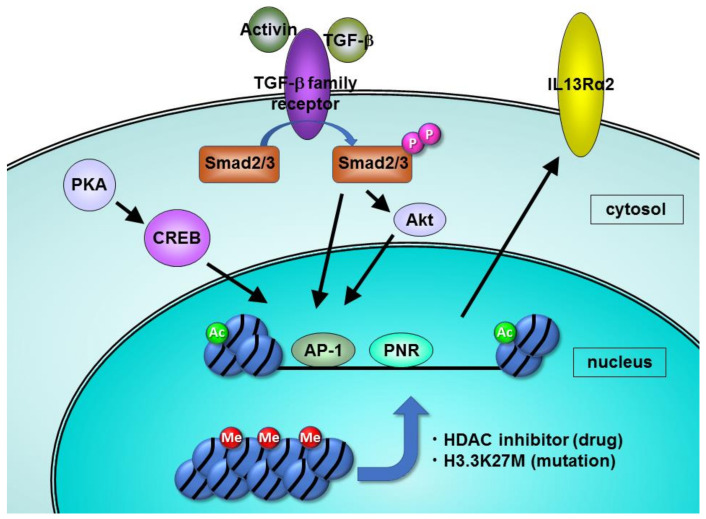
The schema of mechanisms increasing expression of IL-13Rα2 in cancer cells.

**Table 1 diagnostics-11-01140-t001:** The function of IL-13Rα2 and the control of the expression in pancreatic cancer.

Function of IL-13Rα2	Activate ERK 1/2, c-jun, and c-fos
	Increase transcription of MMPs
	Promote tumor invasion and metastasis
	Associated to perineural invasion
	Shorten patient’ survival time
Control of Expression	Histone acetylation enables IL-13Rα2 expression
	c-jun increases transcription of IL-13Rα2

## Data Availability

This article does not report any data.

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
