# Peer review of "IL-13Rα2 Is a Biomarker of Diagnosis and Therapeutic Response in Human Pancreatic Cancer"

_diagnostics, 2021, doi:10.3390/diagnostics11071140_

Round 1
Reviewer 1 Report
Fujisawa et al. discuss the role of IL-13Rα2 expression in pancreatic cancer in this review. The topic is timely and IL-13Rα2 seems a promising marker for PDAC diagnosis and a potential therapeutic target. However, the manuscript needs to be restructured to make it suitable for publication. The role of IL-13Rα2 expression specifically in PDAC needs to be addressed and a critical review of its diagnostic value and accuracy has to be added. Several major and minor aspects should be addressed.
Major
- Please express the expression of IL-13Rα2 thoughout cancer types in more details. Is it heterogeneous? Do expression levels vary within a cancer entity or across different cancer entities?
- How can IL-13Rα2 be implied to diagnose pancreatic cancer if it is expressed in several different cancer entities?
- Can you comment on sensitivity and specificity of IL-13Rα2 expression for PDAC detection in clinical samples?
- Can you outline the differences in detecting IL-13Rα2 expression via IHC vs. ISH, is one more sensitive than the other?
- Is there intra-tumoral heterogeneity of IL-13Rα2 expression? That would be an obstacle for FNA-derived diagnosis.
- Are there known or potential side-effects of targeting IL-13Rα2?
- Is IL-13Rα2 expressed in cancer cells or in the stroma?
- Please reformulate the “future directions” section a bit more critically. What additional benefit could be derived from an IL-13Rα2 PET? Could IL-13Rα2 expression be useful as serum markers for liquid biopsies purposes? What studies are still needed to explore it more as a marker?
Minor
- Please let a native speaker review the manuscript!
Author Response
Fujisawa et al. discuss the role of IL-13Rα2 expression in pancreatic cancer in this review. The topic is timely and IL-13Rα2 seems a promising marker for PDAC diagnosis and a potential therapeutic target. However, the manuscript needs to be restructured to make it suitable for publication. The role of IL-13Rα2 expression specifically in PDAC needs to be addressed and a critical review of its diagnostic value and accuracy has to be added. Several major and minor aspects should be addressed.
Response: We thank the reviewers valuable comments. We have incorporated their changes and revised our manuscript. We believe that the quality of the manuscript is improved after inclusion of all comments/suggestions made by the review team.
Major
- Please express the expression of IL-13Rα2 throughout cancer types in more details. Is it heterogeneous? Do expression levels vary within a cancer entity or across different cancer entities?
Response: Based on published literature and our own research, it has been reported that IL-13Rα2 is highly expressed in pancreatic cancer, glioblastoma, ovarian cancer, breast cancer, colorectal cancer, prostate cancer, melanoma, renal cell carcinoma, hepatocellular carcinoma, pheochromocytoma and thyroid cancer. The expression of IL-13Ra2 vary among different cancer type and clinical stage e.g. more than 95 % of pheochromocytoma, 66% of colorectal cancer, 62% of glioma, 50% of gastric cancer, and 7.5% of melanoma show high expression of IL-13Rα2. The expression of IL-13Rα2 is heterogeneous among the tumor type, grade, and clinical stage. Interestingly, minimal heterogeneity is observed in the pancreatic cancer. We have revised the manuscript in light of these comments ( p.2 l.61).
- How can IL-13Rα2 be implied to diagnose pancreatic cancer if it is expressed in several different cancer entities?
Response: Based on our study, we believe that a validated and well optimized PET/CT technique, may add to other technologies for early diagnosis of pancreatic cancer. In addition, analysis for IL-13Rα2 in clinical samples obtained with EUS-FNA at the time of diagnosis of pancreatic cancer may help assess invasive potential of pancreatic cancer and help predict the patient survival prior to surgery or chemotherapy.
- Can you comment on sensitivity and specificity of IL-13Rα2 expression for PDAC detection in clinical samples?
Response: Immunohistochemistry (IHC) is currently a gold standard for the evaluation of the expression of antigens and receptors in clinical samples including IL-13Rα2 expression in PDAC. Although IHC results were confirmed by in-situ hybridization (ISH) technique, it remained quite challenging to determine sensitivity and specificity of IHC as this technique is qualitative. We used isotype control antibodies for specificity and counted % positive cells in each sample for sensitivity in a blinded manner. Our future aim is to include additional methods/technologies such as real time-PCR and ELISA to corroborate IHC/ISH results for additional sensitivity.
- Can you outline the differences in detecting IL-13Rα2 expression via IHC vs. ISH, is one more sensitive than the other?
Response: We evaluated the expression of IL-13Rα2 in PDA by IHC and ISH techniques as reported in our recent publication (Fujisawa et al. Cancers. 2020; 12.; ref. 25). IHC detected IL-13Ra2 protein expression in paraffin embedded sections by immunostaining with highly specific antibody. ISH detected the mRNA transcripts present in the PDA sections by using highly specific labeled anti-sense riboprobe. Both techniques demonstrated expression of IL-13Ra2 protein and mRNA transcripts, and these results corroborated with each other. No significant difference was observed in the expression of IL-13Rα2 by these two methods. Based on our studies, we are unable to conclude at this time which technique is more sensitive. Additional extensive studies are needed to determine whether IHC is better than ISH or vice versa.
- Is there intra-tumoral heterogeneity of IL-13Rα2 expression? That would be an obstacle for FNA-derived diagnosis.
Response: As mentioned in previous comment that IL-13Ra2 expression is heterogenous in most cancers. However, PDA samples showed least heterogeneity. Because of the intra-tumoral heterogeneity in expression, it is possible one will miss samples by FNA. Therefore, it is recommended to sample at various locations of tumors. We are currently investigating whether the expression of IL-13Rα2 can be confirmed preoperatively using EUS-FNA samples. These results suggest that the heterogeneity within the tumor is not large, and that EUS-FNA can also assess the predominance of IL-13Rα2 expression in the tumors.
- Are there known or potential side-effects of targeting IL-13Rα2?
Response: The results of a phase III study using IL-13Ra2-targted IL-13-Pseudomonas exotoxin local administration to the brain after resection of glioblastoma have been published. Various side-effects were observed, which included aphasia, hemiparesis, deep vein thrombosis etc. However, these side effects were not different from a comparator arm in subjects that received gliadel wafers (Kunwar et al. Neuro-Oncology 2010).
Moreover, Liu-Chittenden et al. performed a phase I clinical trial of systemic intravenous infusion of IL-13-Pseudomonas exotoxin in patients with metastatic adrenocortical carcinoma (Cancer medicine 2015). They reported various side-effects of the immunotoxin, which included anemia, proteinuria, increased creatinine, acute kidney injury, hyponatremia, neutropenia, pain, and thrombocytopenia. At a Maximum tolerated dose, none of these side effects were observed.
- Is IL-13Rα2 expressed in cancer cells or in the stroma?
Response: IL-13Rα2 is expressed specifically in cancer cells. The status of IL-13Ra2 expression in tumor stroma has not been investigated.
- Please reformulate the “future directions” section a bit more critically. What additional benefit could be derived from an IL-13Rα2 PET? Could IL-13Rα2 expression be useful as serum markers for liquid biopsies purposes? What studies are still needed to explore it more as a marker?
Response: We appreciate reviewers’ excellent suggestions. PET/CT targeting IL-13Rα2 may be useful not only for the diagnosis of pancreatic cancer but also evaluating the expression level of IL-13Rα2 and predicting the efficacy of therapy targeting IL-13Rα2. In addition, extracellular domain of IL-13Rα2 may be cleaved and shed in the systemic circulations. Therefore, serum marker by liquid biopsies could be useful as potential biomarker of diagnosis and prognosis and possibly predicting the efficacy of therapy targeting IL-13Rα2. Animal studies and clinical studies are needed to determine whether IL-13Ra2 is secreted in the serum and does it correlate with the clinical outcome. Somatostatin receptor scintigraphy is used not only for the qualitative diagnosis of pancreatic neuroendocrine tumors but also for the pre-treatment evaluation of peptide receptor radionuclide therapy by evaluating somatostatin receptor expression. In response to this comment, we revised the manuscript by adding more information.
Minor
- Please let a native speaker review the manuscript!
Response: Thank you for your suggestions. The manuscript has been reviewed and revised.

Reviewer 2 Report
In their work author performed a comprehensive review on interleukin-13 receptor alpha2 in pancreatic cancer.
The paper is well written and clear to read
Some points need to be addressed
Could you add a figure to resume the role of IL13ra2 in pancreatic cancer?
I would like to add a brief paragraph in the diagnostic section about the prognostic value of IL13ra2 the following reference could be helpful.
Do you think that the interaction of IL13ra with Fibroblast could play a significant role in pancreatic cancer and its resistance to standard therapies?
Future perspectives section should be implemented. the following reference could be helpful. doi: 10.1038/s41598-019-39018-3. doi: 10.1016/j.intimp.2020.107155.
Author Response
The paper is well written and clear to read
Some points need to be addressed
Response: We thank the reviewer for their valuable comments. We have incorporated the changes in the revised version of our manuscript.
- Could you add a figure to resume the role of IL13ra2 in pancreatic cancer?
Response: We added a Table 1 to delineate the role of IL-13Rα2 in pancreatic cancer.
- I would like to add a brief paragraph in the diagnostic section about the prognostic value of IL13ra2 the following reference could be helpful.
Response: As suggested by the reviewer, we added the following sentences in the diagnosis section. “We evaluated the expression of IL-13Rα2 in surgically resected pancreatic cancer and determined any association with post-operative patient prognosis. There were more postoperative tumor recurrences, and the survival was significantly shortened in the group with high IL-13Rα2 expression. Although these differences in expression levels with histology have not yet been analyzed on a large set of clinical specimens, IL-13Rα2 expression may help in the diagnosis of pancreatic cancer and in the prediction of the patient prognosis.”
- Do you think that the interaction of IL13ra with Fibroblast could play a significant role in pancreatic cancer and its resistance to standard therapies?
Response: Thank you for a very insightful comment. It is possible that tumor associated fibroblasts play an important role in pancreatic cancer and its resistance to standard therapy. This is an important area of research we plan to investigate in the immediate future.
- Future perspectives section should be implemented. the following reference could be helpful. doi: 10.1038/s41598-019-39018-3. doi: 10.1016/j.intimp.2020.107155.
Response: As followed reviewer’s suggestion, we have included the following sentences in the diagnosis section. ”Raza et al. developed a novel anti- IL-13Rα2 antibody and examined its antitumor effect in a mouse model of lung cancer34. As a result, anti- IL-13Rα2 antibody reduced tumor metastasis and prolonged the prognosis. In addition to this antibody, a combinatorial approach of IL-13 neutralizing monoclonal antibody35 and blocking peptide to IL-13Rα236, have been investigated for antitumor effects in mouse models. This combined approach of antibodies and peptide therapy to target IL-13Rα2 may have potential in treating human subjects in the future and may be useful in clinical practice. Okamoto et al. found that the expression of amphiregulin, which is a member of the epidermal growth factor (EGF) family positively correlated with IL-13Rα2 expression and promoted tumor growth via activating amphiregulin in melanoma. It is thus possible to target molecules downstream of IL-13Rα2 without directly targeting IL-13Rα2.

Reviewer 3 Report
The authors report a descriptive review about the role of IL-13Rα2 as a biomarker of diagnosis and therapeutic response in pancreatic cancer.The manuscript is complete and well-structured.
Author Response
The authors report a descriptive review about the role of IL-13Rα2 as a biomarker of diagnosis and therapeutic response in pancreatic cancer. The manuscript is complete and well-structured.
Response: We thank the reviewer for his review and useful comments.

Round 2
Reviewer 2 Report
Authors addressed the request satisfactorily. Paper could be accepted